# Signaling Pathways Leading to mTOR Activation Downstream Cytokine Receptors in Lymphocytes in Health and Disease

**DOI:** 10.3390/ijms241612736

**Published:** 2023-08-13

**Authors:** Lucie Fallone, Thierry Walzer, Antoine Marçais

**Affiliations:** CIRI—Centre International de Recherche en Infectiologie (Team Lyacts), Inserm, U1111, Université Claude Bernard Lyon 1, CNRS, UMR5308, ENS de Lyon, F-69007 Lyon, France; lucie.fallone@inserm.fr (L.F.); thierry.walzer@inserm.fr (T.W.)

**Keywords:** mTOR, CD8+ T cell, NK cell, cytokine signaling

## Abstract

CD8+ T cells and Natural Killer (NK) cells are cytotoxic lymphocytes important in the response to intracellular pathogens and cancer. Their activity depends on the integration of a large set of intracellular and environmental cues, including antigenic signals, cytokine stimulation and nutrient availability. This integration is achieved by signaling hubs, such as the mechanistic target of rapamycin (mTOR). mTOR is a conserved protein kinase that controls cellular growth and metabolism in eukaryotic cells and, therefore, is essential for lymphocyte development and maturation. However, our current understanding of mTOR signaling comes mostly from studies performed in transformed cell lines, which constitute a poor model for comprehending metabolic pathway regulation. Therefore, it is only quite recently that the regulation of mTOR in primary cells has been assessed. Here, we review the signaling pathways leading to mTOR activation in CD8+ T and NK cells, focusing on activation by cytokines. We also discuss how this knowledge can contribute to immunotherapy development, particularly for cancer treatment.

## 1. Introduction

The target of rapamycin (TOR) pathway orchestrates metabolic regulation from yeast to plants and humans. More specifically, the activation of the TOR kinase, the effector of the TOR pathway, promotes anabolic pathways, such as translation or synthesis of nucleotides and lipids, while it represses catabolic activities, such as autophagy. TOR kinase activity is regulated by intracellular (e.g., ATP) and extracellular cues (e.g., amino acids) so that anabolic pathways are activated when the building blocks are present while catabolic activities are switched on in less prosperous conditions. In multicellular organisms, however, the organism’s homeostasis maintains nutrient availability [1], thus calling for a second control mechanism to prevent unrestricted growth, a hallmark of cancer [2]. During evolution, extracellular signals controlling cell activation and growth, referred to as growth factors, have thus been co-opted as inputs regulating the TOR signaling network. This explains why mechanistic TOR (mTOR) behaves as an integrator, being active only when both nutrients and growth signals are present. Importantly, the surface receptors that detect such growth signals are different from one cell type to another to ensure the independent regulation of different cell types. As a consequence, the classes of receptors involved in growth signal detection are varied, ranging from tyrosine kinase receptors to G-protein coupled receptors. This diversity suggests that the signaling pathways leading to mTOR activation can differ from one cell type to another and even inside a given cell type if two distinct receptors are triggered. Historically, the study of mTOR signaling has been conducted in vitro in transformed cell lines using the pathway triggered by insulin as a model. This pathway and its components are conserved in most cell types, making it a good model system. However, it does not reveal the whole breadth of the signaling network regulating mTOR activity. In addition, the complete rewiring of cellular metabolism in cultured transformed cell lines makes them a poor model for specific regulatory steps taking place in primary resting cells. Here, we will review the signaling pathways leading to mTOR activation in lymphocytes, taking the example of CD8+ T cells and Natural Killer (NK) cells. Indeed, various steps of these cells’ biology are regulated by cytokines, a subset of growth factors classically associated with hematopoietic or immune cells development, which stimulate the mTOR pathway. We also focused this review on mTORC1 activation, the complex primarily involved in metabolic regulation, since we and others have demonstrated the importance of this complex in the regulation of lymphocyte functions and activity.

## 2. CD8+ T and NK Cell Development and Functions: Common Themes and Divergences

CD8+ T lymphocytes and Natural Killer (NK) cells are cytotoxic lymphocytes specialized in the response to intracellular pathogens and cancer. They are able to kill infected or tumor cells through directed exocytosis of granules containing perforin and granzyme, and produce large quantities of pro-inflammatory cytokines, such as IFN-γ [3]. These capacities place them at the forefront of both anti-tumor and antiviral responses. However, they diverge in the conditions that lead to the acquisition of these effector functions: while naïve CD8+ T cells require clonal expansion and subsequent differentiation into cytotoxic T lymphocytes (CTL) to acquire these capacities, NK cells acquire them during their development. This translates into very different response kinetics during immune responses: while NK cells display their effector functions in a matter of hours, CD8+ T cells require a differentiation step of several days. Moreover, the signals leading to this acquisition also differ. Indeed, naïve T cell differentiation into CTL is triggered upon the engagement of the T cell receptor (TCR), and this first signal is complemented by the help of Interleukin (IL)-2, secreted by helper CD4+ T cells or, to a lesser extent, by CD8+ T cells themselves. In contrast, NK cell development and acquisition of effector potential is principally driven by the cytokine IL-15. IL-2 and IL-15 bind to partially overlapping receptors comprised of IL2Rβ and the γ chain for the common part, while the alpha chains IL2Rα and IL15Rα bring the specificity. Since the α chains are unable to signal, the signal arising from IL-2 or IL-15 receptors is almost indistinguishable [4]. Other cytokines, such as cytokines of the IL-1 family, also take part in the activation of certain functions of CD8+ and NK cells. Indeed, cytokines from the IL-1 family are important for terminal differentiation and acquisition of effector functions, such as IL-1β, which increases CD8+ T cell cytotoxic capacity [5], or IL-18, which, in association with IL-12, is the main driver of IFN-γ production by both CD8+ T cells and NK cells [6]. In addition, it becomes clear that additional factors, such as the presence of metabolites in the extracellular milieu or the bioenergetic state of the cell, strongly influence lymphocyte fate decisions. Therefore, lymphocyte activation depends on the integration of a large set of intracellular and environmental cues, including antigenic signal, cytokine stimulation and nutrient availability. This integration is achieved by signaling hubs, such as mTORC1.

## 3. A Brief Overview of the mTOR Pathway

### 3.1. mTOR Complexes and Targets

mTOR was identified thanks to its sensitivity to rapamycin, a macrolide compound isolated from the bacteria *Streptomyces hygroscopicus* found in a soil sample of Easter Island (Rapa Nui being the native name of the island). It is the core kinase of two complexes, the mTOR complexes 1 (mTORC1) and 2 (mTORC2). mTORC1 is composed of the kinase mTOR associated with mammalian lethal with SEC13 protein 8 (mLST8) and the scaffold regulatory-associated protein of mTOR (RAPTOR). Two endogenous inhibitors also belong to the complex, the proline-rich AKT substrate of 40 kDa (PRAS40) and the DEP domain-containing mTOR-interacting protein (DEPTOR). mTORC2 is organized around mTOR, the protein mLST8 and the scaffold protein rapamycin-insensitive companion of mammalian target of rapamycin (RICTOR) in lieu of RAPTOR, along with DEPTOR (as in mTORC1), mammalian stress-activated protein kinase-interacting protein 1 (mSIN1) and protein observed with RICTOR 1 and 2 (PROTOR ½). The two complexes differ by their sensitivity to rapamycin, mTORC1 being sensitive and mTORC2 not being directly sensitive to it [7], but also by their substrates and functions. mTORC1 plays a well-known role in the control of protein synthesis through the phosphorylation of 4E-BP and S6K, which are classical hallmarks of mTORC1 activation. In addition, mTORC1 also activates glycolysis, lipids and nucleotide synthesis and represses autophagy, therefore supporting anabolism and energy synthesis. Therefore, mTORC1 activation is usually associated with cell growth and proliferation. The role of mTORC2 is less understood; it phosphorylates AKT on Ser473, thereby potentiating its kinase activity, which represses transcription factors FOXO1/3 in lymphocytes [8]. Such factors regulate the expression of multiple genes associated with lymphocyte quiescence or trafficking. Moreover, in some systems, mTORC2 regulates cytoskeleton dynamics. As stated above, this review will mainly discuss mTORC1 signaling and control.

### 3.2. The Importance of mTOR in Lymphocytes

The notion that mTORC1 activity is essential for lymphocyte activation and functions stems from earlier studies that discovered the immunosuppressive activity of rapamycin [9]. mTORC1 inhibitors have thus been used in clinics, notably in the context of lymphoproliferative diseases [10,11,12] or organ transplantation [13,14]. Therefore, the essential role of mTOR for immune cell development and function is now widely recognized, especially in lymphocytes [15]. Immune signals delivered by the TCR and co-stimulatory molecules in T cells, NK-activating receptors in NK cells or cytokine receptors in both T and NK cells trigger mTORC1 activation, which switches on the metabolic pathways required to support cellular activation. mTOR also has a crucial role during NK cell development by controlling their maturation and maintaining their reactivity [16,17]. Furthermore, memory CD8+ T cell differentiation also relies on tight regulation of mTOR activity [18,19,20].

### 3.3. mTOR Upstream Regulation

In eukaryotic cells, mTORC1 senses two broad categories of signals: growth factors and nutrients. The concomitance of these two signals is required for mTORC1 activation as it regulates both mTORC1 kinase activity and localization (Figure 1).

#### 3.3.1. Linking Growth Factors to mTOR Activation: The Example of Insulin Signaling

Growth factors, which we will refer to here as molecules capable of stimulating cell proliferation and survival, are a first set of signals that trigger mTORC1 activation. In 1998, insulin was found to activate mTOR [21] and has since become the best-documented example of mTOR activation by a growth factor. The insulin receptor is a receptor tyrosine kinase expressed by virtually all cells. Insulin binding to its receptor triggers receptor autophosphorylation on several cytoplasmic tyrosines, driving the recruitment and phosphorylation of the adaptor protein Insulin Receptor Substrate (IRS). Mammals express four IRS isoforms, the main ones being IRS1 and IRS2 [22]. Phosphorylated IRSs then serve as binding sites to recruit proteins with SH2 domains, such as phosphatidylinositol 3-kinases (PI3K). Class I PI3K is formed by a 110 kDa catalytic subunit (p110) among the isoforms α, β, γ and δ, associated with a regulatory subunit among p85α, p85β or p55γ. The four p110 catalytic subunits can act upstream of mTOR; however, they are recruited downstream of different receptors, p110γ only being recruited downstream of G-protein-coupled receptors. In mature lymphocytes, the most expressed subunit is p110δ; it associates with p85α or p85β [23]. Class I PI3K catalyzes the phosphorylation of phosphatidylinositol (4,5) biphosphate (PI(4,5)P2) into PI(3,4,5)P3, which acts as a docking site for proteins containing pleckstrin homology (PH) domains, such as the protein kinase B (PKB also named AKT) and 3-phosphoinositide-dependent protein kinase-1 (PDK1). This allows PDK1 to phosphorylate the activation loop of AKT at Thr-308, an essential event for AKT activation. In addition, AKT is also phosphorylated on Ser-473, primarily by mTORC2, even though this modification can be added by other kinases such as TBK1 [24,25,26]. This second phosphorylation event stabilizes the Thr308 phosphorylation and increases AKT kinase activity. IRS also recruits the adaptor GRB2, which binds SOS, a guanine nucleotide exchange factor for RAS. Stimulation of RAS constitutes the starting point of the activation of the ERK mitogen-activated protein kinases (MAPK) cascade.

Regarding the mTOR pathway, a major convergence point of the AKT and ERK pathways is the inhibition of the tuberous sclerosis complex (TSC), a key inhibitor of mTORC1. TSC is composed of three proteins, TSC1, TSC2 and TBC1D7 [27], and acts as a GTPase-activating protein (GAP) for the small GTPase RHEB [28,29], an essential mTORC1 activator. By inhibiting TSC, growth factors such as insulin thus allow mTOR to be fully activated by RHEB. Structural studies show that RHEB loaded with GTP binds to mTOR distally from the kinase site and causes a global conformational change of the complex [30]. This molecular mechanism explains that GTP-loaded RHEB behaves as an allosteric activator increasing mTORC1 activity by several orders of magnitude. On note, RHEB has unusually slow intrinsic GTPase activity. As a result, TSC, which stimulates RHEB GTPase activity and therefore inhibits this small GTPase, behaves as a gatekeeper of mTORC1 activity. In addition to this well-described canonical pathway, some studies also suggest that the mTORC1 complex itself can be post-translationally modified following growth factor stimulation [31,32]. The relevance of such modifications is unclear.

The activity of TSC is tightly regulated by multiple proteins. AKT phosphorylates TSC2 at different sites, thus suppressing its control of RHEB and mTOR [33]. TSC is also inhibited by phosphorylation from ERK [34] and the p90 ribosomal S6 kinase (RSK) [35], another downstream kinase of the Ras-ERK pathway. Other studies described a regulation of TSC by growth factors through the control of its recruitment to the lysosome [36,37,38,39]. The regulation of TSC through the disruption of the complex itself or through its degradation is still debated (reviewed in [40]). Overall, these results define TSC as a major hub collecting inputs that negatively regulate mTORC1.

#### 3.3.2. mTOR Control by Nutrients and Cellular Stress

In addition to growth factors, nutrients are essential for mTORC1 activation. In particular, the role of amino acids has long been recognized and involves a set of four GTPases, RAG GTPases A, B, C and D, that act as heterodimers combining RAG A or B and RAG C or D [41,42]. In conditions of amino acids sufficiency, the RAG GTPases are locked in an “on-state”, with RAG A/B bound to GTP and RAG C/D bound to GDP, and vice versa in case of amino acid scarcity. In their “on-state”, RAG GTPases mediate mTORC1 attachment to the lysosome surface via a multi-proteic complex called the Ragulator [43,44], bringing mTORC1 in close vicinity to its activator RHEB, which also lies on the external membrane of the lysosome. Thus, when nutrients are abundant, mTORC1 is tethered to the lysosome surface, where it can be fully activated by mitogen signals, such as insulin or other growth factors. On the contrary, when nutrients are limiting, mTORC1 is released from the lysosome surface and thus inactivated. This two-step mechanism explains the importance of the coincidence of both nutrients and growth signals for mTORC1 activation and supports its role as a signal integrator. Several proteins are involved in amino-acid sensing both inside the lysosome and the cytoplasm and constitute today a topic of active research (review in [45]). Of note, not all amino acids seem to have the same potency. Arginine, Leucine, and Methionine have been described as direct regulators of mTORC1, each one possessing its dedicated sensor [46]. Other nutrients, such as glucose [47] or cholesterol [48,49,50], have also been shown to impinge directly onto the mTORC1 pathway in a manner partially overlapping the amino acid pathway.

In addition to this direct control, the lack of nutrients can also be sensed via the decrease of ATP. Indeed, energy drop or cellular stress also negatively regulate mTOR activity. Glucose depletion activates the AMP-activated protein kinase (AMPK), which senses the decrease in ATP/ADP ratio and inhibits mTORC1 either directly through phosphorylation of RAPTOR on inhibitory sites [51] or indirectly by activating the phosphorylation of TSC2 [52]. Independently of AMPK, hypoxia also negatively regulates mTORC1 through TSC activation [53,54]. Several mTORC1 negative regulators, including TSC or AMPK, are targets of p53, which plays a central role in DNA damage sensing [55]. Finally, endoplasmic reticulum stress or oxidative stress also negatively regulate mTORC1 function (review in [56]).

## 4. mTOR Activation Downstream of γc Cytokine Receptors in Lymphocytes

Cytokines are soluble proteins with signaling functions in the immune system, and some of them have a growth factor activity. Physiologically, γ chain cytokines, such as IL-2 and IL-15, are potent mitogens for T and NK cells. IL-1 family cytokines increase cytotoxic functions and cytokine production of lymphocytes, two processes that require energy. Therefore, it is no surprise that these cytokines activate the mTORC1 pathway. The next two sections discuss our current knowledge of mTORC1 regulation in T and NK cells, focusing on γ chain and IL-1 type cytokines.

### 4.1. Overview of the γc Cytokines Family

The γ chain (γc) family regroups cytokines that signal through the common receptor γ chain, namely IL-2, IL-4, IL-7, IL-9, IL-15 and IL-21. γc was first discovered as a component of the IL-2 receptor, making IL-2 the archetypal member of this family [57,58]. The IL-2 receptor is composed of three proteins: CD25, the α chain of the IL2R (IL-2Rα); CD122, the β chain of the IL2R (IL-2Rβ); and the γ chain CD132. As mentioned before, IL-15 signals through two of the three IL-2 receptor chains, γc and IL-2Rβ, while its unique IL-15Rα chain serves to transpresent IL-15 from neighboring cells [59,60]. In terms of signaling, most of our knowledge comes from studies on the IL-2 receptor. Contrary to the insulin receptor, the IL-2 receptor lacks intrinsic kinase activity. IL-2Rβ and γc are associated with the kinases JAK1 and JAK3, respectively, and the binding of IL-2 results in JAK1/3 activation and phosphorylation of tyrosine residues on IL-2Rβ [61,62,63]. These residues serve as docking sites for proteins containing SH2 domains. Phosphorylated IL-2Rβ binds the p85 subunit of PI3K [64], which triggers the activation of the PI3K-AKT pathway. Phosphorylated IL-2Rβ also serves as a docking site for the GRB2-SOS complex through the adaptor SHC, which constitutes the starting point of the Ras-ERK MAPK cascade. The transcription factor STAT5 is also recruited to the activated IL-2R and is of particular importance for IL-2 signaling [65].

### 4.2. mTOR Regulation by γc Cytokines in T Cells

The first evidence of mTOR activation by γc cytokines comes from studies using IL-2 [66,67]. In T cells, IL-2 treatment triggers ribosomal protein S6 phosphorylation, a marker of S6K activity downstream of mTORC1. IL-2 treatment is also associated with an increase in cell size and metabolism, other indirect hallmarks of mTORC1 activation [68,69,70,71]. Several studies show an increase in AKT phosphorylation a few minutes after IL-2 treatment, confirming that IL-2 activates this pathway in T cells [69,70,72]. However, the importance of PI3K-AKT in mTORC1 activation in T cells is debated. Studies from Cantrell’s group show that the inhibition of AKT or PI3Kδ does not decrease mTORC1 activity following IL-2 treatment in CTL [69]. The same group, however, showed that PDK1 is required for mTORC1 activity, since CTL deleted for this protein display decreased S6 phosphorylation, glucose uptake and cell proliferation [68]. Conversely, the deletion of PTEN, a negative regulator of PI3K signaling that can drive the activation of AKT, is not sufficient to activate mTORC1 and mTORC1-dependent glucose metabolism [73]. This evidence suggests that the PI3K-AKT pathway is activated by IL-2 but is dispensable for mTORC1 activation. How PDK1 controls mTORC1 independently of the PI3K-AKT pathway remains to be deciphered. One hypothesis would be that its activity is not required extemporaneously but more upstream to phosphorylate and license the neosynthesized S6K [74].

How can IL-2 activate mTORC1 if not through the well-described PI3K-AKT axis? One possibility could be through the JAK-STAT pathway. STAT5-deficient CD4+ T cells treated with IL-2 show decreased mTORC1 activity, as measured by S6 phosphorylation, associated with a reduction in cell size, proliferation and metabolism compared to their WT counterpart. However, STAT5-deficient T cells have also decreased levels of AKT phosphorylation at Thr308, suggesting that STAT5 may act upstream from AKT. To directly test the relationship between STAT5 and AKT, T cells deleted for STAT5 were complemented with either constitutively activated STAT5 or constitutively activated AKT. Only STAT5 transduction rescued mTORC1 activation, whereas the transduction of AKT did not, suggesting a mechanism whereby mTOR activation relies on STAT5 while it is independent of AKT [70]. To explain the activation of AKT, Ross and Cantrell proposed a mechanism involving pathways depending on the SRC family kinases LCK and FYN. This signaling, independent of STAT5, would participate in the context of “preorganized” signal transduction pathways, which integrates with IL-2-STAT5 signaling [72]. On note, STAT5 is a transcription factor and activates the transcription of several key proteins relevant for mTORC1 signaling, such as IL-2R chains, AKT, RHEB or amino acid transporters and sensors, which participate in longer-term activation of mTORC1 [70,71].

In T cells, the γc cytokine IL-7 is also known to activate mTORC1, although the molecular details are less documented than for IL-2. Indeed, IL-7 treatment enhances S6 phosphorylation [75] as well as cell size and survival [76] and glucose uptake [77]. This last metabolic change seems to require AKT and STAT5 activation [77], although no clear evidence has demonstrated a molecular link between these pathways and mTORC1 in IL-7-treated lymphocytes.

### 4.3. mTOR Regulation by γc Cytokines in NK Cells

NK cell maturation and survival is governed by the γc cytokine IL-15. IL-15 treatment activates mTORC1, which increases cellular metabolism [16]. IL-15 also activates STAT5, AKT [16], ERK [78] and its downstream target RSK [79]. Several studies have investigated the role of these different proteins in the activation of mTORC1. Using pharmacological inhibitors, Nandagopal and colleagues showed that PI3K or STAT5 inhibition following IL-15 treatment decreased NK cell production of IFN-γ and granzyme B as well as its proliferative capacity to the same extent as rapamycin. However, the status of mTORC1 activation itself was not measured in this study [80]. NK cells deficient in PTEN, an antagonist of PI3K, display increased AKT and S6 phosphorylation [81], while NK cells deficient in the catalytic subunits of PI3K p110γ and p110δ display a defective maturation and function [82]. Interestingly, the observed defect included a strong decrease in circulating NK cells and a nearly complete disappearance of the most mature CD11b+ subset, phenocopies of mTOR or RAPTOR deficiencies [16,83,84]. In addition, IL-15 activates the ERK pathway. The use of an MEK inhibitor suggests that ERK could participate in mTORC1 activation in NK cells, as ERK inhibition leads to decreased S6 phosphorylation following IL-15 treatment [78]. Overall, these data are indicative of a role of both the PI3K-AKT and ERK pathways in the control of mTORC1 in NK cells. On the contrary, a study conducted in IL-2/-12 in vitro-activated NK cells reached the opposite conclusion that the PI3K-AKT pathway is not involved in mTORC1 activation. Indeed, in this last study, NK cells treated with AKT inhibitors show no decrease in cell size, IFN-γ production, granzyme expression or S6 phosphorylation [85]. This suggests that mTORC1 regulation differs when comparing CD8+ T cells and resting or activated NK cells and that their regulation can be adapted to the cellular activation status. The reason for these differences is unknown at this point (Figure 2).

### 4.4. The Role of the TSC Axis in Lymphocytes

As described above, TSC constitutes a hub controlling mTORC1 activation thanks to its capacity to integrate both negative and positive inputs. As a result, TSC null cell lines show maximal mTORC1 activity to the same extent as insulin-treated ones [86,87], whereas TSC overexpression results in drastic decreases in mTORC1 activity [33,52,88,89]. A similar high activity of mTORC1 is observed in vivo in primary muscle cells [90] or in hepatocytes [91] upon TSC inactivation. For these reasons, different groups looked at the importance of the TSC hub in T and NK cells.

Four distinct studies have investigated the role of TSC in T cells and concluded that TSC is required to maintain cell quiescence in the absence of antigen stimulation [20,92,93,94]. TSC1-specific deletion in immature T cells, using the T cell-specific CD4-Cre deleter, leads to a decrease in CD4+ and CD8+ T cell numbers [92,93,94]. The remaining T cells show an increase in cell proliferation and display an activation of the intrinsic apoptotic pathway [93,94], characterized by a decrease in the anti-apoptotic protein Bcl-2 [93]. They also display elevated cellular ROS, which could be reversed by antioxidant treatments that restore T cell numbers [93,94]. This phenotype is associated with an increase in mTORC1 activity and a decrease in mTORC2 activity [93,94]. Such a decrease in mTORC2 activity is supposed to arise from the negative feedback loop leading from S6K to targets lying upstream of mTORC2 [45]. Focusing on CD8+ T cells, Pollizzi et al. show that TSC2-deficient T cells display characteristics of terminally differentiated effector T cells and are unable to transition to the memory state. Confirming these results, T cells deficient in the GTPase RHEB fail to differentiate into effector cells but retain memory characteristics [20].

In NK cells, TSC1 deletion at an early stage produces a similar phenotype as TSC1 deletion in immature T cells, with a decrease in NK cell number associated with an increase in cell proliferation and apoptosis and a constitutive mTORC1 activation [95]. However, the deletion of TSC1 in mature cells does not have any effect on NK cell phenotype, function or response to cytokines [95]. In the same line, inducible inhibition of TSC1 at latter stages in T cells using an ER-Tamoxifen construct has only a mild effect on the T cell population [93]. On note, TSC is mainly expressed by progenitors, and its expression decreases during the maturation process, both in NK and T cells [95], which could account for the more important phenotype of TSC deletion in immature cells compared to mature ones. These results also echo those from Cantrell’s group, suggesting that PI3K-AKT signaling is not mandatory for mTOR regulation by γc chain cytokines. In fact, T and NK cell homeostasis mainly depends on the γc cytokines IL-2 and IL-15, respectively, which, as discussed earlier, may not depend on the classical pathway involving AKT and, therefore, TSC, being one of the main targets of AKT, for their signaling in lymphocyte. On note, even if these studies show the importance of TSC in the maintenance of cell quiescence, TSC1 mutations in lymphocytes result in a mild phenotype compared to other cell types. For instance, mice bearing liver-specific TSC1 ablation developed hepatocellular carcinoma [91]. This suggests other mechanisms of regulation involving TSC-independent control of mTOR in mature lymphocytes.

## 5. mTOR Activation Downstream of IL-1β Family Receptor in Lymphocytes

### 5.1. Overview of the IL-1β Cytokines Family

The IL-1 family is a large group of cytokines composed of agonist molecules, such as IL-1α, IL-1β, IL-18, IL-33, IL-36α, IL-36β, IL-36γ and IL-37, and antagonist ones, such as IL-1Rα. They all signal through receptors composed of two different chains, which are all characterized by the presence of a conserved cytosolic region containing a signaling domain called the Toll/IL-1 Receptor (TIR) domain. The binding of the cytokine to the first receptor chain triggers the recruitment of the second one and the juxtaposition of their TIR domains. This induces the recruitment of the protein MyD88 and the kinase IRAK4. The latter is activated by autophosphorylation, leading to the subsequent recruitment and phosphorylation of IRAK1 and 2, two adaptors and kinase proteins. MyD88 and its IRAK partners then form a protein platform that recruits the E3 ubiquitin ligase TRAF6, which triggers the activation of different pathways, such as the MAPK and NF-κB pathways [96].

IL-1 family cytokines typically engage the MAPK p38. There are four isoforms of p38 (p38α, β, γ, δ), the most expressed in mammals being p38α. As for other MAPKs, p38 activation results from phosphorylation cascades involving different MAP3Ks, such as TAK1, which is classically engaged by IL-1. Furthermore, as described for other MAPKs, p38 has the potential to phosphorylate a multitude of targets, including other kinases such as the MAPK-activated protein kinases (MAPKAPK) MK2, MK3 and MK5/PRAK [97]. Several studies report the control of mTORC1 by p38 in different cell lines, mentioning both an activator [98,99,100,101] or inhibitor [102] role for p38 depending on the stimulus involved. In addition to the p38 MAPK pathway, IL-1 family cytokines also engage the NF-κB pathway. NF-κB activation requires activation of the IKK complex through its binding to polyubiquitin chains on several molecules downstream of the IL-1 receptor, such as IRAK1 or TAK1. The activated IKK phosphorylates IκBα, which promotes its degradation and the release and nuclear translocation of NF-κB, which promotes the pathway activation. IKK has also been involved in mTORC1 activation. Indeed, in one study, cancer cell lines treated with TNFα activate IKKβ, a component of the IKK complex that interacts with and phosphorylates TSC1, resulting in TSC1 inhibition and mTORC1 activation [103].

### 5.2. mTOR Regulation by IL-1β Cytokine Family in T and NK Cells

In T and NK cells, different cytokines of the IL-1 family have been shown to activate mTORC1, although the molecular pathways involved are not fully understood yet. mTORC1 has been shown to be activated by IL-18 [16,104] or by IL-33 [105] in NK cells, and by IL-1 in Th17 [106,107] and IL-36b in CD8+ T cells [108,109]. The most complete mechanistic study comes from Gulen et al. [107]. Working on Th17, the authors showed that IL-1 induces the activation of mTORC1, measured by S6 and 4EBP1 phosphorylation. The authors also demonstrated that IL-1 signals through the IRAK1 and IRAK4 proteins to induce TSC disruption. The precise mechanism of TSC inactivation is still yet to be described. The authors proposed that inactivation of this complex is the event leading to mTORC1 activation. In the same study, IL-1 treatment is also shown to induce p38, JNK and IκBα phosphorylation, although the implication of these pathways in TSC disruption was not clearly addressed [107] (Figure 3). Another study demonstrated that T cells deficient in p38α/β or MK2/3 show decreased mTORC1 activity as measured by S6 phosphorylation upon TCR stimulation [110]. This defect correlates with increased regulatory T cell (Treg) numbers in vivo, which is reminiscent of the phenotype of mTOR-deficient T cells [111]. This suggests that p38 could control mTORC1 activation in lymphocytes; however, the significance of this pathway for cytokine signaling remains to be determined. As mentioned earlier, mTORC1 activity is also controlled by nutrient availability. In this context, Almutairi et al. showed that IL-18 treatment leads to enhanced surface expression of the leucine transporter CD98/LAT-1 on NK cells. This resulted in increased leucine import, thus probably participating in mTORC1 activation [104]. It should, however, be noted that CD98 induction by IL-18 takes place after an overnight stimulation; it is thus unlikely that this mechanism is at play in short-term (<4 h) activation of mTORC1 upon IL-18 stimulation.

### 5.3. Signal Integration toward mTOR Control

Multiple signaling pathways modulate mTORC1 activity, as reviewed here. The outcome of co-stimulation with multiple cytokines will have to be deciphered. Indeed, we mostly focused our review on the effect of cytokines taken in isolation. However, cells generally encounter cytokines simultaneously. For instance, the generation of the so-called CIML (cytokine-induced memory-like cells) for therapeutic purposes implies the stimulation of NK cells with IL-15 and IL-18, both having the potential to activate mTORC1 with parallel pathways. Moreover, coincident stimulation with different cytokines can arguably lead to contradictory effects. For instance, T cell and NK cell proliferation and cytotoxicity induced in vitro with IL-2 can be inhibited by TGF-β treatment [112,113]. Interestingly, TGF-β antagonizes the IL-15 activation of mTOR in NK cells, resulting in a reduction in NK cell bioenergetic metabolism. This inhibition is detectable in the minutes following TGF-β treatment and is specific to mTOR since it does not impact STAT5 activation [114]. More recently, TGF-β has been shown to decrease mTOR activity in T cells [115]. In this study, early inhibition of mTOR in precursors of exhausted CD8+ T cells preserved their metabolic capacity and sustained the T cell response during chronic infection. In both studies, the molecular mechanism at play remains elusive but could well involve some of the molecular players described above. Indeed, TGF-β has been shown to potentially activate a large set of pathways in a variety of cell types. This includes TAK1 [116], ERK [117], p38 [118] and PI3K [119]. Another possible mechanism to explain the inhibition of mTORC1 by TGF-β is the phosphatase PP2A. PP2A has been shown to inhibit mTORC1 in Tregs, and the ablation of PP2A in Tregs leads to severe autoimmunity [120]. The direct link between PP2A and the TGF-β signaling still remains to be determined. Whether activation of these pathways enters in competition with their involvement in mTOR signaling remains to be tested.

Some studies also highlight that several transduction pathways can be engaged for a given stimulus. For instance, in NK cells, a low dose of IL-15 only triggers STAT5 activation and cell survival, whereas higher doses engage mTORC1 and cell proliferation [16]. Similarly, in another model studying stress granule formation upon arsenite treatment, Heberle et al. addressed the respective importance of PI3K-AKT and p38 pathways in the control of mTORC1 activation in multiple cell lines. Using both in silico and in vitro approaches, the authors showed that PI3K-AKT and p38 pathways act in a hierarchical manner toward mTORC1 activation. Inhibition of PI3K results in residual mTORC1 activation, measured by p70-S6K phosphorylation, which is inhibited by an additional treatment with a p38 inhibitor. Based on these findings, the authors further show that the effect of p38 inhibition on mTORC1 activity is significant when PI3K activity is decreased to 40% or lower. From these results, it appears that moderate stress activates mTORC1 through p38, whereas increasing the stress signal also engages the PI3K-AKT pathway, leading to stronger mTORC1 activation [121]. Even though this study has not been performed on lymphocytes, we can hypothesize that similar mechanisms may take place in T and NK cells in response to different cytokine concentrations.

## 6. Therapeutic Targeting of mTOR

### 6.1. mTOR Inhibition in Cancer and Autoimmunity

As mTORC1 is essential for lymphocyte activation and function, the dysregulation of mTOR activity is associated with several lymphocyte defects leading to pathologies such as malignancies and autoimmune disorders. T cell acute lymphoblastic leukemia (T-ALL) is frequently characterized by activating mutation of NOTCH1 or loss-of-function mutation of PTEN [122,123]. Interestingly, these two mutations result in mTORC1 activation, and therefore, mTOR inhibitors have been proposed in T-ALL treatment. In a PTEN-deficient mouse model, rapamycin halts T-ALL initiation and development. However, most of these mice still developed T-ALL after rapamycin withdrawal. The author concluded that rapamycin effectively eliminates leukemia blast but fails to inhibit mTOR in leukemia stem cells, which could contribute to the limited clinical efficacy of rapamycin in T-ALL [124].

In parallel, mTOR is hyperactivated in up to 80% of cancers [45], founding the use of mTOR inhibitors as antiproliferative drugs in chemotherapies in other cancer types. This is not without side effects. Indeed, studies from our group have shown that patients with metastatic breast cancer treated with the mTOR inhibitor everolimus display lower levels of mTORC1 activation in NK cells as well as a decrease in NK cell number and maturation [125]. As NK cells have crucial antimetastatic activity [126], this side effect of mTOR inhibitors should be considered for the development of next-generation therapies. At the same time, mTOR inhibition may also benefit certain types of responses and promote certain differentiation pathways. Indeed, as discussed above, mTORC1 inhibition favors memory CD8+ T cell generation [18,20] and has a protective effect against T CD8+ exhaustion in chronic infection [115]. For these reasons, mTORC1 activation level in immune cells has to be well balanced to ensure a favorable anti-tumor effect.

Overactivation of the mTOR pathway in T cells is also associated with autoimmune disorders such as systemic lupus erythematosus [127]. mTOR inhibitors have been shown to reduce autoimmunity symptoms in both a mouse model [128] and in humans [129]. Indeed, in patients with systemic lupus erythematosus, rapamycin treatment has been shown to improve the disease, associated with an increase of Treg and CD8+ memory T cells and a decrease of pro-inflammatory CD4+ and CD8+ T cells [130].

### 6.2. Targeting mTOR Activation in Lymphocytes for Immunotherapy

Controlling mTOR signaling can also constitute a promising strategy to modulate lymphocyte activity. In this context, a large set of therapies are now aiming at boosting lymphocyte functions, in particular for adoptive cell transfer therapies in cancer. Chimeric Antigen Receptor (CAR) T and NK cell-based immunotherapies have revolutionized the treatment of certain cancers [131]. The use of such therapies to treat viral infections (EBV and others) is also under assessment [132]. A better understanding of cytokine signaling toward mTOR could lead to improved T and NK cell expansion in vitro. The study of cytokine signaling also has the potential to benefit in tailoring better cytokine cocktails in order to prime lymphocytes in vitro. In the same line of idea, a better understanding of the signal transduction pathways involved in downstream cytokine receptors could inform the design of engineered receptors to improve CAR cell effector functions (review in [133]). Therefore, mTORC1 tight regulation is required to ensure a proper response to pathogens and cancer, which will benefit from a better understanding of its specific regulation in immune cells.

## 7. Concluding Remarks

The importance of the kinase mTOR for lymphocyte development and function is now widely recognized. Our knowledge of mTOR signaling comes mostly from early studies on the transformed cell line. If these studies allowed the discovery of the major molecular actors involved in mTOR control, such as the PI3K-AKT pathway or the TSC-Rheb axis, their relative importance in primary cells is now under debate. Therefore, further work focusing on mTOR regulation in primary cells should benefit our understanding of lymphocyte regulation, and could find a use in therapy design.

## Figures and Tables

**Figure 1 ijms-24-12736-f001:**
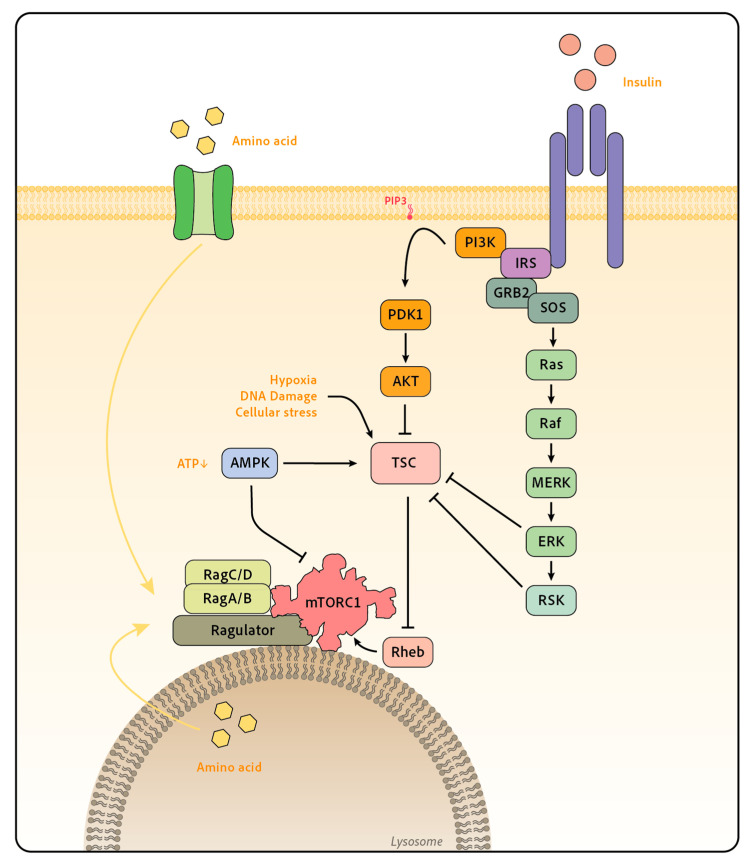
Overview of mTORC1 upstream regulation. mTORC1 activation is controlled by both growth factors and nutrients. Growth factors such as insulin activate both the PI3K-AKT signaling and the MAPK ERK pathway, which converge on TSC inhibition. TSC acts as a GAP for RHEB, which boosts mTORC1 kinase activity. Among nutrients, amino acids are essential for mTORC1 activation as they control mTORC1 translocation to the lysosome through the activation of the RAG GTPase A/B and C/D. At the lysosome, mTORC1 is in close vicinity to its activator RHEB. This two-step mechanism allows mTORC1 to integrate both growth factors and nutrient presence. Other parameters, such as energy level or hypoxia, further regulate mTORC1.

**Figure 2 ijms-24-12736-f002:**
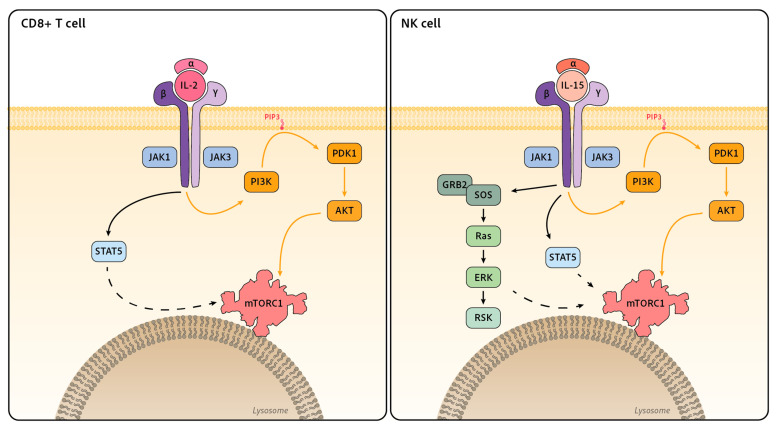
Signaling downstream of IL-2 and IL-15 receptors toward mTORC1 in CD8+ T cell and NK cells. IL-2 and IL-15 receptors share both their IL-2Rβ and γ chain (γc) associated with a unique IL-2Rα or IL-15Rα chain depending on the receptor. IL-2Rβ and γc are associated with the kinases JAK1 and JAK3, respectively. The binding of the cytokine results in JAK1/3 activation and phosphorylation of tyrosine residues on IL-2Rβ, which serves as an activating platform for STAT5, PI3K and GRB2-SOS. Even though the PI3K-AKT pathway is engaged by the IL-2 receptor (in orange), its importance in mTORC1 activation remains under debate.

**Figure 3 ijms-24-12736-f003:**
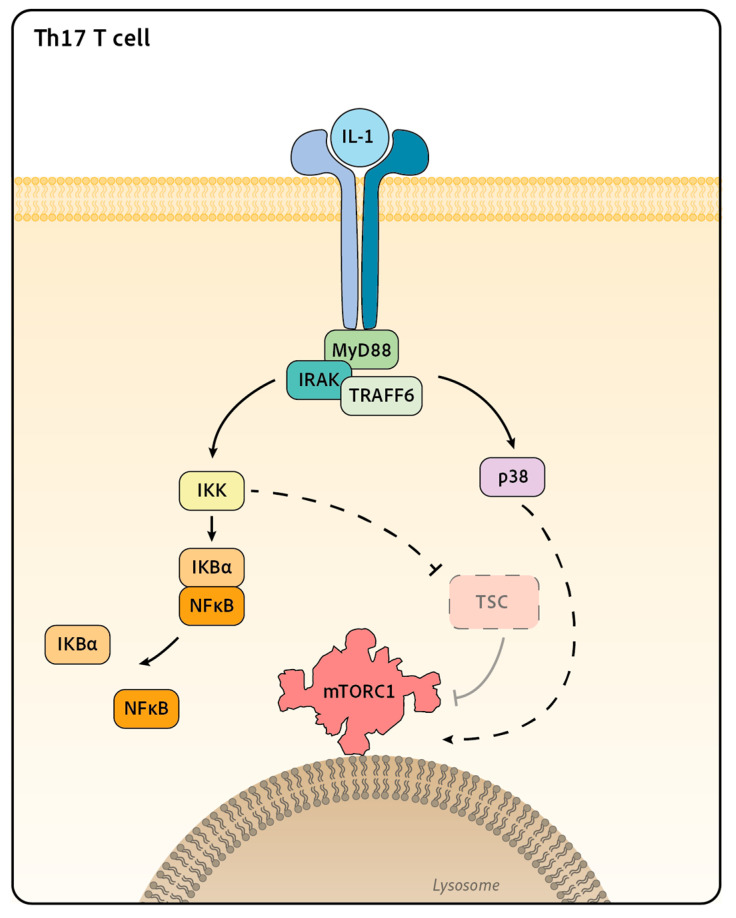
Signaling downstream of IL-1 receptor toward mTORC1 in T cell. IL-1 binding to its receptors induces the recruitment of the adaptors MyD88 and IRAKs. This activates the ubiquitin E3 ligase TRAF6, which triggers the activation of MAPK, such as p38 and the NFκB pathways. In Th17 T cell, Gulen et al. showed that IL-1 treatment drives the disruption of TSC and the activation of mTORC1. The role of the MAPK p38 or NFκB in this process in lymphocytes still remains to be determined, as p38 and IKK have been shown to activate mTOR in other cell types.

## Data Availability

Not applicable.

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
