# Peer review of "Signaling Pathways Leading to mTOR Activation Downstream Cytokine Receptors in Lymphocytes in Health and Disease"

_ijms, 2023, doi:10.3390/ijms241612736_

Round 1

Reviewer 1 Report

I read this review with great interest.

I found it well written and accepted in the current form. 

Author Response

To the Editors of “International Journal of Molecular Sciences”,

Lyon, 8th August 2023

Dear Editors,

We would like to thank the reviewers for this very fair and thorough analysis of our manuscript. We have made several changes in the manuscript (underlined in yellow) to follow reviewers’ recommendations. In addition, we addressed reviewers’ comments point-by-point; our response is presented below. We hope that you will find the manuscript now suitable for publication in the International Journal of Molecular Sciences.

All authors concur with the submission.

Looking forward to hearing from you,

Sincerely yours,

Reviewer 1:

I read this review with great interest.

I found it well written and accepted in the current form.

We thank this reviewer for his positive appreciation of this work.

Reviewer 2 Report

This review represents a comprehensive summary of the activation of mTOR signaling in CD8 T and NK cells by different signals.

Major point: The title of the review states "in health and disease" (and the special issue is regarding mTOR in human diseases) but there is only one short paragraph about the mTOR as a therapeutic target. The authors should add a paragraph to elaborate more on the role of mTOR in cancer and anti-viral response.

The text is nicely structured and very detailed on the different activators and downstream players of mTOR, so the reviewer suggests minor changes regarding phrasing or sentence construction for easier reading. 

1. lines 27-30: sentence too long, shorten "...when the required building blocks are present, while catabolic activities are switched-on in less prosperous conditions."

2. lines 35-36: the phrase coincidence detector is used in neurobiology with different meaning, which is not appropriate here, please revise. 

3. lines 104-105: Therefore, mTORC1 activation is usually associated with cell growth and proliferation.

4. line 118: co-stimulatory, line 429: co-stimulation

5. lines 201-202: remove "content resulting from it" from the text

6. lines 268-274: quotes needed

7. line 301: order of words "observed defect"

8. line 430: remove "of the literature on data reporting"

Authors should revise the use of commas everywhere in the text as well as prepositions (eg. on the contrary instead of IN the contrary, of note instead of ON note, to the same extent as instead of to the same extent THAT etc.)

Author Response

Reviewer 2:

We would like to thank this reviewer for his feedbacks. We modified the manuscript by taking into account the remarks made. The detail of the modifications is listed below.

This review represents a comprehensive summary of the activation of mTOR signaling in CD8 T and NK cells by different signals.

Major point: The title of the review states "in health and disease" (and the special issue is regarding mTOR in human diseases) but there is only one short paragraph about the mTOR as a therapeutic target. The authors should add a paragraph to elaborate more on the role of mTOR in cancer and anti-viral response. We added a section on the role of mTOR in diseases (paragraph 6.1) This section reviews mTOR dysregulations in lymphocytes in cancer (T-ALL) and autoimmune disorders (lupus) and therefore elaborates on mTOR dysregulation in lymphocyte in diseases.

The text is nicely structured and very detailed on the different activators and downstream players of mTOR, so the reviewer suggests minor changes regarding phrasing or sentence construction for easier reading. We thank the reviewer for his positive assessment.

  1. lines 27-30: sentence too long, shorten "...when the required building blocks are present, while catabolic activities are switched-on in less prosperous conditions." This sentence has been changed into: “TOR kinase activity is regulated by intracellular (e.g. ATP) and extracellular cues (e.g. amino acids), so that anabolic pathways are activated when the building blocks are present while catabolic activities are switched-on in less prosperous conditions.”
  2. lines 35-36: the phrase coincidence detector is used in neurobiology with different meaning, which is not appropriate here, please revise. We replaced “coincidence detector” by “integrator”.
  3. lines 104-105: Therefore, mTORC1 activation is usually associated with cell growth and proliferation. The sentence “Altogether, this explains that mTORC1 activation is usually associated with cell growth and proliferation” has been changed into “Therefore, mTORC1 activation is usually associated with cell growth and proliferation.”
  4. line 118: co-stimulatory, line 429: co-stimulation. Costimulation has been changed to co-stimulation
  5. lines 201-202: remove "content resulting from it" from the text. This sentence has been modified as follows: “In addition to this direct control, the lack of nutrients can also be sensed via the decrease of ATP.”
  6. lines 268-274: quotes needed How can IL-2 activate mTORC1 if not through the well described PI3K-AKT axis? This sentence is a rhetorical question and therefore does not need quotes to the best of our knowledge.
  7. line 301: order of words "observed defect" The phrase the “defect observed” has been changed into the “observed defect”
  8. line 430: remove "of the literature on data reporting" The sentence “Indeed, we mostly focused our review of the literature on data reporting the effect of cytokines taken in isolation.” has been changed into “Indeed, we mostly focused our review on the effect of cytokines taken in isolation.”

 Authors should revise the use of commas everywhere in the text as well as prepositions (eg. on the contrary instead of IN the contrary, of note instead of ON note, to the same extent as instead of to the same extent THAT etc.) “ON the contrary” has been change to “IN the contrary”, “OF note” has been change to “ON note”, and “to the same extent THAN” to “to the same extent THAT”. The use of commas has been revised too.

Reviewer 3 Report

Ref: ijms-2474776
Title: Signaling pathways leading to mTOR activation downstream cytokine receptors in lymphocytes in health and disease.
Journal: IJMS-MDPI

The manuscript entitled: “Signaling pathways leading to mTOR activation downstream cytokine receptors in lymphocytes in health and disease” by Fallone et al.is a well written article reviewing the signaling pathways leading to mTOR activation in CD8 T and NK cells, with a focus on activation by cytokines. The article fits within the scopus of the journal. Therefore, I recommend this manuscript to be accepted for publication in the IJMS journal, after some minor revisions (please see below):

 Abstract: Line 18: “Discussed” should be replaced with “discuss”; Line 19:” development for intracellular pathogen”: please rephrase to reflect more on the impact of cancer here.

-Line 51:” cytokines, a subgroup of growth factors”: please elaborate whether a cytokine could be also a growth factor and maybe rephrase to make this clearer.

-Line 75: “cytokines of the IL-1 family”: please specify the members of this family. Are the members of only IL-1 important? If there are others please mention them.

-Line 327:” similar derepression of mTORC1 is observed..”: please rephrase since the word de-repression is not very clear.

-Line 468:” Targeting mTOR activation in lymphocytes for immunotherapy development”: the word development could be substituted or removed.

-“Targeting mTOR activation in lymphocytes for immunotherapy development”: It would be preferable here to expand more on the impact of cytokines and involvement as immunotherapeutic method.

-A summary table of the impact of mTOR activation downstream cytokine receptors in lymphocytes in different diseases would be useful to be added.

-A conclusion section would be useful to be included.

English language is fine; only minor edits are required.

Author Response

Reviewer 3:

The manuscript entitled: “Signaling pathways leading to mTOR activation downstream cytokine receptors in lymphocytes in health and disease” by Fallone et al.is a well written article reviewing the signaling pathways leading to mTOR activation in CD8 T and NK cells, with a focus on activation by cytokines. The article fits within the scopus of the journal. Therefore, I recommend this manuscript to be accepted for publication in the IJMS journal, after some minor revisions (please see below):

We thank the reviewer for his positive words of appreciation.

 Abstract: Line 18: “Discussed” should be replaced with “discuss”; We modified “We also discussed” for “we also discuss”.

Line 19:” development for intracellular pathogen”: please rephrase to reflect more on the impact of cancer here. We change this sentence into “we also discuss how this knowledge can contribute to immunotherapy development, particularly for cancer treatment” to emphasize more on the impact for cancer therapy.

-Line 51:” cytokines, a subgroup of growth factors”: please elaborate whether a cytokine could be also a growth factor and maybe rephrase to make this clearer.

The following changes have been made to make the definition clearer: line 35-36 “During evolution, extracellular signals controlling cell activation and growth, referred to as growth factors, have thus been co-opted as inputs regulating TOR signaling network” and line 60-62: “Indeed, various steps of these cells’ biology are regulated by cytokines, a subset of growth factors classically associated with hematopoietic or immune cells development, that stimulate the mTOR pathway.”

-Line 75: “cytokines of the IL-1 family”: please specify the members of this family. Are the members of only IL-1 important? If there are others please mention them. The IL-1 family members are mentioned line 366-367 in the sentence “The IL-1 family is a large group of cytokines composed of agonist molecules such as IL-1a, IL-1b, IL-18, IL-33, IL-36a, IL-36b, IL-36g, IL-37 and antagonist ones like IL-1Ra.”

-Line 327:” similar derepression of mTORC1 is observed..”: please rephrase since the word de-repression is not very clear. “Derepression” has been change to “A similar high activity of mTORC1”

-Line 468:” Targeting mTOR activation in lymphocytes for immunotherapy development”: the word development could be substituted or removed. “Development” has been removed

-“Targeting mTOR activation in lymphocytes for immunotherapy development”: It would be preferable here to expand more on the impact of cytokines and involvement as immunotherapeutic method. We feel that this development is beyond the scope of our review.

-A summary table of the impact of mTOR activation downstream cytokine receptors in lymphocytes in different diseases would be useful to be added. Instead of a table, we added a section on mTOR dysregulation in lymphocytes in diseases (section 6.1). In this paragraph, we discuss mTOR activation in lymphoma (T-ALL) and autoimmune disorder (lupus).

-A conclusion section would be useful to be included. We added a “Concluding remarks” section (section 7).

English language is fine; only minor edits are required. We revised the English grammar and spelling.